# Copper-Polyurethane Composite Materials: Particle Size Effect on the Physical-Chemical and Antibacterial Properties

**DOI:** 10.3390/polym12091934

**Published:** 2020-08-27

**Authors:** Cristian Miranda, Johanna Castaño, Emky Valdebenito-Rolack, Felipe Sanhueza, Rody Toro, Helia Bello-Toledo, Patricio Uarac, Luciano Saez

**Affiliations:** 1Unidad de Desarrollo Tecnológico (UDT), Universidad de Concepción, Coronel 41919960, Chile; fesanhueza@udec.cl; 2Facultad de Ingeniería y Tecnología, Universidad San Sebastian, Lientur 1457, Concepción 4080871, Chile; rody.toro@uss.cl (R.T.); patricio.uarac@uss.cl (P.U.); luciano.saez@uss.cl (L.S.); 3Laboratorio de Investigación de Agentes Antimicrobianos (LIAA), Departamento de Microbiología, Facultad de Ciencias Biológicas, Universidad de Concepción, Concepción 4080871, Chile; emkyvaldebenito@udec.cl (E.V.-R.); hbello@udec.cl (H.B.-T.)

**Keywords:** copper particle, polyurethane, particle size, composite, antibacterial

## Abstract

In this work, thermoplastic polyurethane (TPU) composites incorporated with 1.0 wt% Cu particles were synthesized by the melt blending method. The effect of the incorporated copper particle size on the antibacterial, thermal, rheological, and mechanical properties of TPU was investigated. The obtained results showed that (i) the addition of copper particles increased the thermal and mechanical properties because they acted as co-stabilizers of polyurethane (PU) (ii) copper nanoparticles decreased the viscosity of composite melts, and (iii) microparticles > 0.5 µm had a tendency to easily increase the maximum torque and formation of agglomerates. SEM micrographics showed that a good mixture between TPU and copper particles was obtained by the extrusion process. Additionally, copper-TPU composite materials effectively inhibited the growth of the Gram-negative *Escherichia coli* and the Gram-positive *Staphylococcus aureus*. Considering that the natural concentration of copper in the blood is in the range of 0.7–0.12 mg/L and that the total migration value of copper particles from TPU was 1000 times lower, the results suggested that TPU nanocomposites could be adequately employed for biomedical applications without a risk of contamination.

## 1. Introduction

Advances in nanotechnology as a multidisciplinary field represent a revolutionary path for the development of functional materials with unique physical and chemical properties for numerous applications [1]. The production of polymer systems containing micro- and nanostructures may provide solutions for modern medicine and engineering, as well as agro-food, textile, and food technologies, among others. Therefore, more attention has been focused on the processing and stability of functional materials with higher performances. Nanostructures exhibit a high reactivity and increased tendency for atomic and ion exchange. Therefore, the incorporation of nanostructures into the polymeric matrix can impart new properties to composite materials [1]. Metal nanostructures, such as copper and silver nanoparticles, are of particular interest, since they exhibit biological and antibacterial activity and can be successfully used in medicine, agriculture, and catalysis [2,3]. Jain and Prapeed, 2005, prepared polyurethane composite foams by soaking samples in a silver nanoparticle solution. They demonstrated the antibacterial properties of the silver-coated polyurethane foam against *Escherichia coli* and suggested the use of this material as a drinking water filter. Additionally, a total reduction in adhered *Escherichia coli* on PVC/Cu composites containing 0.5 wt Cu nanoparticles was reported by Becerra et al., 2013 [4]. These materials were prepared by the solving casting method.

The highly rough surfaces of polymeric material allow the adhesion of bacteria, forming biofilms, which represent the main defense mechanism of these microorganisms against environmental agents. Biofilms are organized communities of bacteria attached to surfaces embedded in a matrix of extracellular polymeric substances (EPS), in which the cells are protected from a hostile environment [5]. There are clear differences between the physiology of a biofilm and a planktonic bacterial community, for example, a microorganism’s biofilm has a higher resistance against antibiotics; microorganisms do not display substrate-limited growth, because they interact with the environment; biofilms are not fixed structures; and a bacteria’s biofilm produces EPS [6]. The observation of biofilms by scanning electron microscopy (SEM) shows that the EPS are filaments connected to bacterial cells and the material surface, because of the dehydration process occurring prior to the SEM [7]. It is estimated that 80% of prokaryotes are in their environment in a biofilm form [6]. One of the most promising strategies for the elimination of biofilms is to avoid bacterial adhesion on the surface of the material, through the function of plastic materials that have built-in antibacterial sustained release, such as metal nanostructures with microbial properties, which avoid surface colonization on the rough surfaces of the polymeric matrix used, for example, in the application of films for agriculture, food packaging, and medical devices, among others [4]. 

In the biomedical industry, polyurethanes are known as biocompatible polymers that have attractive properties, such as strength, flexibility, elasticity, high processability, and versatility [8]. Polyurethanes (PUs) are used in almost all fields of human activity. Their characteristics or versatility depend on polyurethane chemistry, which is based on the exothermic reaction between di- or poly-isocyanates and compounds with hydroxyl end-groups, such as polyols [9]. Both thermoplastics and thermosets can be formed from PU groups. Thermoplastic polyurethane (TPU) is a linear, segmented copolymer consisting of alternating hard segments (HSs) and soft segments (SSs) [10].

A few studies on TPU with copper micro- and nanostructures can be found in the literature [8]. They are focused on obtaining biologically active polyurethane materials that contain silver (30–90 nm) and cooper (10–140 nm) by solvent evaporation. 

The purpose of this paper is to obtain copper-thermoplastic polyurethane composites containing different sized copper particles by the melt blending method and to study their physical-chemical properties and antibacterial activity for potential biomedical applications.

## 2. Materials and Methods 

### 2.1. Materials

Thermoplastic polyurethane pellets under the trade name TPU Desmopan^®^, Bayer, Germany, were supplied by Mathiesen-Chile, Santiago, Chile. Copper nanostructures (15 and 20 nm) were obtained from Ion copper LTDA, and microstructures (0.05, 0.2, 0.5, 1, 5, and 10 μm) were obtained from CEDENNA, Santiago, Chile. 

### 2.2. Preparation of Copper-Polyurethane Composite Materials

The compositions of the samples and their codes are summarized in Table 1. A Brabender internal mixer (Plastograph^®^ EC plus, Mixer 50EHT32, Duisburg, Germany) was used to prepare copper-polyurethane composites (CPC) with 1.0 wt% of Cu particles. The samples were blended at 160 °C and 60 rpm for 7 min and torque variation was recorded as a function of time. The torque values of the maximum peak and steady state were determined from the torque rheometer data (see Table 1). 

Neat thermoplastic polyurethane and copper-polyurethane composites were compression molded in a hydraulic press (Labtech Engineering M-Scientific, LP20-B, Samutprakarn, Thailand) at 190 °C for 10 min and 180 bar, and then cooled to room temperature under the same pressure. Films of a 500 μm thickness were obtained for microbiology assay, SEM, and mechanical analysis.

### 2.3. Characterization of Copper-Polyurethane Composite Materials

#### 2.3.1. Evaluation of the Processability of Composite Materials by Torque Rheometry and the Melt Flow Index (MFI)

The rheological characterization of thermoplastic polyurethane and copper-polyurethane composites was carried out in an Instron CEAST MF-20 plastomer (Darmstadt, Germany) according to ASTM D1238 for each polyurethane sample with 1% copper of different particle sizes. The MFI was calculated as the mass of the material in grams flowing per 10 min under a constant load of 2615 kg and temperature conditions of 190 °C, according to ASTM D1238.

#### 2.3.2. Thermal Stability

The thermal stability of neat thermoplastic polyurethane and copper-polyurethane composites was evaluated using a NETZSCH 209 F3 TGA Tarsus Selb thermogravimetric analyzer (Selb, Germany). TGA tests were carried out at 10 °C/min, in a temperature range between 30 and 600 °C, under nitrogen atmosphere (10 mL/min).

#### 2.3.3. Scanning Electron Microscopy (SEM) 

The fracture surface, the observation of the bacterial biofilm surface of films of neat thermoplastic polyurethane, and the copper-polyurethane composite were evaluated using a JSM 6380 LV JEOL scanning electron microscope (Tokyo, Japan) equipped with an ISIS INKA L3000QL Oxford energy dispersive spectroscope (EDS) operated at 20 kV. The surfaces were coated with gold films with a 40 nm thickness by employing sputter coater S5150 equipment. Cu mapping of the horizontal surface was performed for 7.5 mm^2^ of the area using EDS to determine the degree of distribution of Cu particles in the TPU matrix. The magnifications were 2000×, 5000×, and 10,000×. The analysis of bacterial biofilms formed by the strains *Escherichia coli* ATCC 25922 and *Staphylococcus aureus* ATCC 25923 was conducted on neat TPU and CPC. Plates of 1 × 1 cm were submerged in 1 mL of the corresponding bacterial suspension at a concentration of 1.5 × 10^6^ CFU mL^−1^, in 24-well plates, and then incubated at 37 °C. Samples of the neat TPU and CPC were extracted at 24, 48, and 72 h. The neat polyurethane molds were used as a positive control for biofilm growth. A neat polyurethane mold submerged in 1 mL of tryptic soy broth (TSB) without bacterial inoculum was used as a negative control. Three replications of each of the experiments were performed. The extracted samples were coated with gold as described previously. The SEM of the gold plated samples for cell attachment observations was conducted at magnifications of 2500× and 10,000×. The biofilm formation was considered when cells were attached to the material surface and exhibited filaments of exopolysaccharides (EPS) and fimbria, which are bacterial structures usually found in these kind of micrographs [11,12].

### 2.4. Antibacterial Properties

#### 2.4.1. Bacterial Strains

The Gram-negative *Escherichia coli* ATCC 25922 and Gram-positive *Staphylococcus aureus* ATCC 25923 bacterial strains, cultured in TSB or agar (TSA), were used for all of the antibacterial experiments in this study.

#### 2.4.2. Death Kinetics

The antibacterial activity of neat thermoplastic polyurethane (TPU) and a copper-polyurethane composite (CPC) was assessed for overnight cultures of the bacterial strains. Plates of 1 × 1 cm of neat TPU and CPC with 1% copper of all particle sizes studied were inoculated with 20 μL of a suspension (1.5 × 10^6^ CFU mL^−1^) of each bacterial strain separately and then incubated at 37 °C in a humid chamber. The bacterial colony-forming units on TSA plates were produced at 2, 4, 6, and 24 h. In all of the experiments, polyurethane molds without copper were used as a control. Three replications of each of the experiments were performed. The cell density was expressed as Log_10_ of the colony-forming units per mL (Log_10_ CFU/mL) and the average values and standard deviations were then calculated and plotted using Prim 7 (GraphPad, San Diego, CA, USA). A bacteriostatic effect was considered to have occurred when the decrease in the inoculum was >2 Log_10_ and <3 Log_10_ and a bactericidal effect was considered to have occurred when the cell density decrease was ≥3 log_10_ [13,14,15].

#### 2.4.3. Statistical Analysis

Independent Student’s t-tests with a confidence interval of 95% (α = 0.05) were used to assess the significance of the difference between the cell density on TPU and the cell density on each PU-copper material, using Prim 7 (GraphPad, San Diego, CA, USA).

### 2.5. Cu^2+^ Migration 

Copper ion migrations from copper-polyurethane composites in aqueous media were determined from composites with a 1%wt copper concentration and compared with neat TPU. A total of 25 g of each sample was immersed in 250 mL of deionized water and physiological serum, and the samples were shaken for 30 min daily in a horizontal rotary shaker at a speed of 30 rpm during the 3 weeks that the experiment lasted. Then, 5 mL of aliquots was taken to quantify the copper concentration in solution and determine the percentage of copper migration into the aqueous medium. The aliquots were taken at 0, 6, 24, 48, and 72 h, and 7, 14, and 21 days. The concentration of copper in aqueous medium was measured by inductively coupled plasma atomic emission spectroscopy (ICP-OES) and the result was delivered in units of mg/L of copper in solution.

### 2.6. Tensile Properties 

Tensile tests of films were performed according to the ASTM D638 standard on a Karg Industrietechnik universal testing machine (Krailling, Germany) using type V dumbbell samples. The crosshead speed was set at 10 mm/min. The tensile modulus, tensile strength, and percent of elongation at break were calculated from the stress–strain curves. At least seven individual measurements were carried out for each film formulation. The energy required to perform tensile testing until rupture was calculated according to Bei et al. [16].

## 3. Results and Discussion

### 3.1. Processability Evaluation of Composite Materials by Torque Rheometry and the Melt Flow Index (MFI)

The processing properties of the pristine polyurethane and copper-polyurethane composite were studied by monitoring the evolution of torque over time during mixing in a Haake rheometer (Table 1). The evaluated samples show similar behavior in terms of the torque variation, where, within the processing time, the torque increases rapidly to the maximum and then decreases gradually until reaching the equilibrium value (steady state) associated with a typical non-Newtonian characteristic of polymer materials [17,18]. The value of the torque in the steady state was obtained from the torque vs. time curve (Figure 1), and corresponds to the state of equilibrium [19,20].

A slight variation in maximum torque could be observed in composite materials compared to neat polyurethane. Therefore, the incorporation of nanostructures (<0.5 µm) tends to decrease the maximum torque, while the incorporation of microstructures (>0.5 µm) shows a tendency to increase it, suggesting that a larger particle size favors the formation of aggregates, thus requiring more energy to melt. The viscosity of the composite materials during melt blending is related to steady state torque. The copper-polyurethane composite exhibits a low steady state value compared to neat thermoplastic polyurethane. However, composite materials with a smaller size of copper particles show the lowest values. This behavior suggests that the interlayer interaction between TPU chains is reduced by spherical nanoparticles acting as a “lubricating effect,” decreasing the viscosity of composite melts. This result is in accordance with the reports by Miranda et al. [21] and Rylski et al. [22]. The MFI values confirm the rheological behavior of composites described above (Table 1). It is known that variation of the viscosity after processing depends on the level of degradation and the type of polymer. However, the incorporation of 1% of copper structures (nano or micro) does not affect the processability of composite materials obtained by industrial processes. It is important to highlight that the torque values corresponding to steady state conditions are almost constant after 5 min, which indicates that no chain degradation occurred during the processing of the samples.

### 3.2. Thermal Stability

The thermal degradation of TPU is a complex heterogeneous process and involves several partial decomposition reactions where random chain scission and crosslinking are predominant routes for the decomposition [23]. The thermal degradation curves of the neat TPU and its copper-polyurethane composite prepared with nanoparticles exhibit similar behavior to the two main steps of decomposition, while the copper-polyurethane composite processed with microparticles displays three steps (see Figure 2). The lost weight remains almost unchanged (1%) up to 200 °C and afterwards, the weight loss rate quickly increases for all samples (Figure 2a). Lower thermal degradation is associated with trapped volatile materials being released.

The maximum decomposition temperature (T_max_) and associated weight loss at each decomposition step depend on the size of copper particles, as can be observed in Table 2. The first marked stage of decomposition occurs between 250 and 370 °C, which is attributed to the breakage of urethane bonds (soft segments) that dissociate into primary amines, or olefins and carbon dioxide [24]. The maximum decomposition temperature in this step is 354 °C for neat TPU and around 366 °C for the copper-polyurethane composite, suggesting that copper nanoparticles act as co-stabilizers of TPU. On the other hand, the incorporation of microparticles in the TPU polymer matrix apparently induces the appearance of a small peak of decomposition before 350 °C with approximately 7% mass loss (Figure 2b). No increase or thermal stability is observed during the initial decomposition.

The second step ranges from 380 to 450 °C, where TPU degradation is basically related to the scission and depolymerization of soft segments (polyols) [25]. The maximum decomposition temperature in this step is 418 °C for pristine TPU and around 421 °C for copper-polyurethane composite processing with nanoparticles and microparticles with a weight loss of 65% and 63%, respectively (Figure 2b). In the third degradation step of the copper-polyurethane composite processed with microparticles, the maximum rate of degradation takes place at around 428 °C, with a weight loss of 64%. The results of the compounds with larger particle sizes suggest that oxidation of the metallic copper CuO occurs at around 290 °C [26], which is evidenced by the increase in thermal stability (10 °C) of these samples in the last decomposition step. The kinetics of the thermal decomposition depend on the electron-acceptor (aromatic) or electron-donor (aliphatic) character of the isocyanate in the copper-polyurethane composite [27]. The copper acts as a strong acceptor of electrons, promoting coupling reductive reactions with the isocyanate and stabilizing the polymer during the stages of thermal degradation. The residue at 600 °C for all samples is around 9% (Figure 2a) and probably belongs to complex char or oxidized organic matter and inorganic substances.

### 3.3. Scanning Electron Microscopy

The fracture surface and elemental composition in the specific regions of the TPU and copper-polyurethane composites were examined by SEM and EDS essays (Figure 3).

The SEM micrographs on neat TPU show a smooth and homogeneous surface. A smooth surface could be a suggestion of an amorphous material [28]. Lei et al., 2017, reported amorphous halos at around 19° and 43.1° in polyurethane elastomer, which were characteristic peaks of soft segments [29]. The copper-polyurethane composite exhibits a continuous phase, and a homogeneous structure is formed. This suggests that a good mixing between TPU and copper particles is obtained by the extrusion process. On the other hand, no aggregation of copper nanoparticles is observed and the images evidence good adhesion between the copper nanofiller and TPU polymer matrix, which is decisive for any functional composite. However, as the particle size increases, it can be seen that efficient incorporation fails, mainly due to filler agglomeration, because microparticle agglomerates have a lower reactive surface than nanoparticles. The EDS spectra show distinctive red bright points homogeneously dispersed in the TPU matrix, indicating the occurrence of a distributed crystalline phase associated with copper particles. The black region corresponds to the organic matrix of TPU, which is generally amorphous and mainly composed of carbon and oxygen [30]. In the neat TPU, copper has not been detected by EDS. These results allow us to conclude the formation of a semi-crystalline composite material.

### 3.4. Antibacterial Properties

#### 3.4.1. Biofilm Formation

The SEM micrographs of the neat polyurethane and copper-polyurethane composite submerged in bacterial cultures are displayed in Figure 4. All of the experiments exhibit bacterial growth in comparison with the negative control, in which no bacteria are observed (Figure 4c). All of the *E. coli* micrographs have a magnification of 5000× and one micrograph at 2500× for the 72 h of incubation is included in each case as a zoomed out image.

*E. coli* growth is retarded in the CPC, showing less bacteria than in the neat TPU, no exopolysaccharide filaments (EPS; red arrows), and no fimbria (green arrows) at 24 h of incubation. This tendency is the same until the end of the experiment at 72 h of incubation, when significantly less bacteria and EPS are exhibited in the copper-polyurethane composite (Figure 4a, 72 h at 5000×), suggesting that adding Cu particles to the polyurethane inhibits biofilm formation.

On the other hand, all of the *S. aureus* micrographs have a magnification of 2500× and one micrograph at 10,000× for 72 h of incubation is included as a zoomed in image. The *S. aureus* growth is similar for the neat thermoplastic polyurethane and the copper-polyurethane composite until 48 h (Figure 4b, 24 and 48 h), but at 72 h, the CPC show significantly less bacteria than in the neat TPU (Figure 4b, 72 h) and no EPS filaments These results suggest that the copper-polyurethane composites inhibit the formation of biofilms of *E. coli* by contact, and a similar result is observed with *S. aureus*, where the effect remains similar, but is only observable after 72 h. (Figure 4b, 72 h 10,000×). In the mentioned Gram-positive bacteria, the inhibitory effect on the biofilm formation appears later, at 72 h. The inhibitory effect of Cu on the bacterial biofilms found in this study is similar to that presented in other previously described research and some mechanisms have already been proposed [31,32]. Recently, Cu nanoparticles (CuNPs) were shown to exhibit powerful anti-biofilm activity on pathogen bacteria such as *Escherichia coli, Salmonella typhi, Shigella flexneri*, and *Pseudomonas aeruginosa*, because cell death was mediated by massive destruction of the plasmatic cell membrane and the production of high concentrations of reactive oxygen species also known as ROS [33]. Similar results have also recently been published in terms of the use of Cu oxide (CuO) NPs [34,35].

#### 3.4.2. Death Kinetics

The bactericidal activity of CPC (different particle sizes) versus the time on the bacterial strains *Escherichia coli* ATCC 25922 (a) and *Staphylococcus aureus* ATCC 25923 (b) is shown in Figure 5. The plotted values are the average of the cell density (bacterial cell count Log_10_ CFU/mL) and its standard deviation from experiments conducted in triplicate.

As can be seen in Figure 5a, PU-Copper (1%) samples of all particle sizes show a fast bactericidal effect against *E. coli* ATCC 25922 (decreasing ≥ 3 Log_10_) after 1 h of incubation, while a neat TPU sample shows a bacteriostatic effect (2 Log_10_ < cell density < 3 Log_10_). These results demonstrate that the polyurethane-copper composite materials evaluated in this work have excellent antibacterial activity, reaching the total bacterial death after a short contact time.

On the other hand, Figure 5b displays a bacteriostatic effect of all the materials studied against *S. aureus* ATCC 25923 (2 Log_10_ < cell density < 3 Log_10_). Only the cell density of *S. aureus* on PU-Cu of a 200 nm particle size exhibits a significant difference with the cell density of the mentioned species on TPU (*p* < 0.05). These findings suggest (along with the bactericidal effect on *E. coli*) that the TPU with copper particles of 200 nm have the potential to prevent infections in general, similar to results previously reported [36]. In the same way, previous research using copper-containing materials demonstrated that *S. aureus* was less sensitive to this metal than *E. coli* [37].

### 3.5. Cu^2+^ Migration

Values of the released copper ions from material in deionized water and physiological serum measured at different times are shown in Table 3.

The presence of copper ions was not detected during the first hours (<detection limit) in deionized water, in contrast with physiological serum media, where migrations started immediately and the copper ions were detected after one hour (0.05 day). Nevertheless, nanocomposites show a small percentage of copper migration into both media, and copper migration in physiological serum media is larger than in deionized water. After 21 days, the values reach 0.00035% and 0.0015% in deionized water and physiological serum, respectively. These migration values are quite low, and are even below the detection limit (0.02 µg/L) during a day in aqueous media.

Considering the potential use of this kind of material for the production of medical devices, another important parameter corresponds to the total (natural) concentration of copper in the blood of 0.7–0.12 mg/L, whereas the total migration values of copper particles are 1000 times lower, generating no risk of possible contamination.

In comparison with our previous work, where it was reported that copper ion migration from PVC-nano copper composites was 0.45 µg/L after 24 h in deionized water [16], the migration value of copper from TPU is even lower. It is important to mention this, because the stability of copper nanoparticles in the thermoplastic matrix reaffirmed the potential that this kind of antibacterial material has for the manufacturing of medical devices. This work shows that the migration of copper ions from the TPU matrix is very low for at least three weeks, practically eliminating the risk of metal contamination.

### 3.6. Tensile Properties

The effect of the copper size particles on the tensile properties of neat TPU and CPC is shown in Figure 6. The tensile properties include the tensile modulus (E), tensile strength (TS), and elongation at break (EB). The mechanical properties of materials are associated with the nature, chemical structure, and interactions among the forming components. The incorporation of copper particles results in an increase in tensile strength (Figure 6a) and elongation at break (Figure 6b) compared to the neat TPU. The highest values of mechanical properties are observed when copper of 0.05 and 0.2 µm is incorporated in the polymer matrix.

The CPC reinforced with microparticles presents lower E (MPa) compared to the other composites and neat TPU (Figure 6c). This can be attributed to the lack of adherence between the matrix and the reinforcement or to the poor distribution of the nanofiller in the matrix by the tendency to form aggregates. The great dielectric constant of the metallic nanoparticles favors the attractions between them. This attraction is stronger at a short range, and falls quickly with increasing separation [21]. Hence, the tendency to form aggregates is higher with increasing the Cu nanoparticle size in the TPU matrix. On the other hand, the incorporation of copper nanoparticles could promote the interconnection of hard elements forming paths, producing a change in the distribution of the compound which translates into an increase in the toughness of the material [38,39]. Copper-polyurethane composites prepared with nanoparticles were found to be more resistant, with an increase of TS of 23% and flexible materials (>70% of EB), than neat TPU (Table 4).

The incorporation of microparticles shows an increase of TS of approximately 12% and EB of 20% with respect to the TPU matrix (Figure 5). There are several reported mechanisms that seek to explain how mechanical properties are increased with the incorporation of nanostructures. Bei et al. [16] associated simultaneous enhancement of the tensile strength, strain at break, and modulus of PP–LDPE-SiO2 composites with an adequate sample preparation method that led to an improvement of the compatibility and the crystallization structures, which was determined by the addition of the filler masterbatch. Makwana et al., explained that good homogeneity, processing conditions, and interactions between nanoparticles and matrix led to an improvement of the mechanical properties of MMT nanocomposites, which allowed the transfer of efforts of the matrix to the nanoparticles [40].

An interesting theory on the non-monotonic behavior of multiplicity fluctuations in heavy-ion collisions has been well-developed by Mrówczyński et al. [41] and Rybczynski et al. [42]. We hypothesized that during the melt mixing process of composites, the metallic nanoparticles that are uncharged when entering the system can be electrostatically charged given the arrangement of the dipole space. The polarization of the dipoles depends on the distance between them and the particle size [43]. The effect of the size and fiber orientation on reinforcement in mechanical properties of polymeric composites was studied by Zu et al. [44]. They found that fibers oriented at 90° exhibited higher values of Young’s modulus than those oriented at 0°. Hence, the latter had a higher failure stress value of 104 MPa in comparison to the composite with fibers oriented at 90°, which had a value of 84 MPa. Higher fiber sizes mostly oriented in the mold flow direction during injection were obtained during the last stage of failure. In our case, one can conclude that under monotonic loading, the predominant non-linear deformation mechanism is observed in TPU composites due to small and spherical nanoparticles offering less resistance and aligning better in the direction of the flow produced in the injector than large and aggregate particles.

For a convenient comparison, and according to Bei et al. [16], the following integral expression of the stress–strain curve was used to describe the energy required to perform tensile tests until rupture:(1)$$E\;=\int_{0}^{\varepsilon b}\tau d\varepsilon ,$$
where τ is the apparent tensile stress, ε is the corresponding tensile strain, and εb is the strain at break. The values obtained for the stress–strain curve according to Equation (1) are presented in Table 4 and Figure 7. The energy values are directly related to the ductility of the materials. Therefore, the materials that present greater values for the energy integral will exhibit higher ductility. The highest calculated value is around 4100 KJ/m^3^, which corresponds to PU-0.05 and PU-0.2 samples.

The results presented in this work show that good behavior can be expected for this material when used for biomedical applications such as catheters, anti-infective materials, solution packaging, or dialysis tubes (Table 4), according to the results reported by Leandro et al., 2005 [45] and Thein-Han, et al., 2009 [46]. Therefore, some of the mechanical requirements requested in the main polymers used in medical applications present a higher flexibility (EB) and resistance (TS): Silicone rubber EB > 220%, TS = 12 [46]; PVC E > 370%, TS = 16 [33]; and this study EB > 500%, TS > 19. In fact, the reported tensile test parameters are similar to the results for the various tubes, and are all within the specification required for biomedical applications (EB > 200% and TS = 15 MPa) [45].

## 4. Conclusions

Copper nanoparticles incorporated into the TPU matrix improved the mechanical and thermal properties, without affecting the processability of composite materials.

The copper particle size influenced the physical, chemical, and antibacterial properties of composites. A larger particle size increased the maximum torque and there was no evidence of any increase or thermal stability during the initial decomposition process. Moreover, the presence of a small peak of decomposition before 350 °C was observed. Samples incorporated with smaller particle sizes of copper showed the highest values for the mechanical properties and an increased thermal stability. SEM micrographs evidenced that the Cu nanoparticles added to the polyurethane inhibited the biofilm formation of *S. aureus and E. coli* in composite materials. The mechanical and thermal properties of nanocomposites were in accordance with the specification required for biomedical applications.

## Figures and Tables

**Figure 1 polymers-12-01934-f001:**
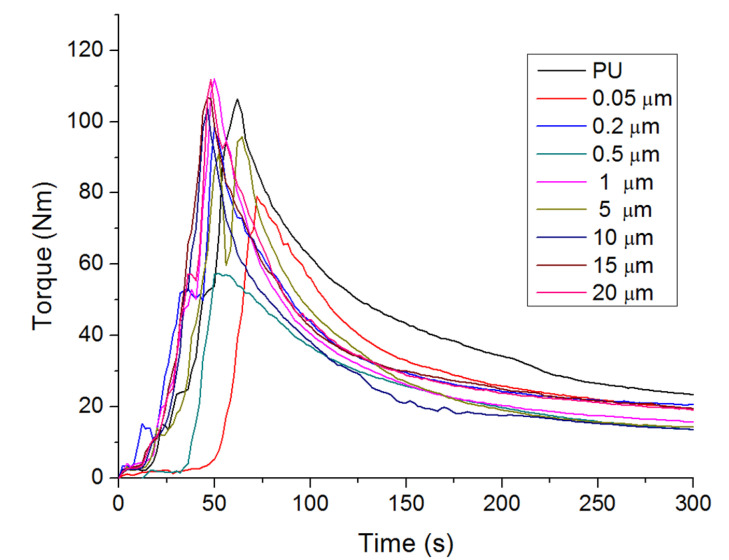
Torque variation as a function of time for samples.

**Figure 2 polymers-12-01934-f002:**
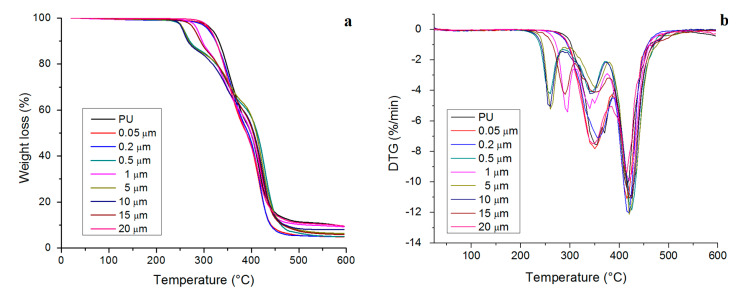
The thermal stability of thermoplastic polyurethane (TPU) and composites evaluated by TGA: (**a**) TGA and (**b**) DTG.

**Figure 3 polymers-12-01934-f003:**
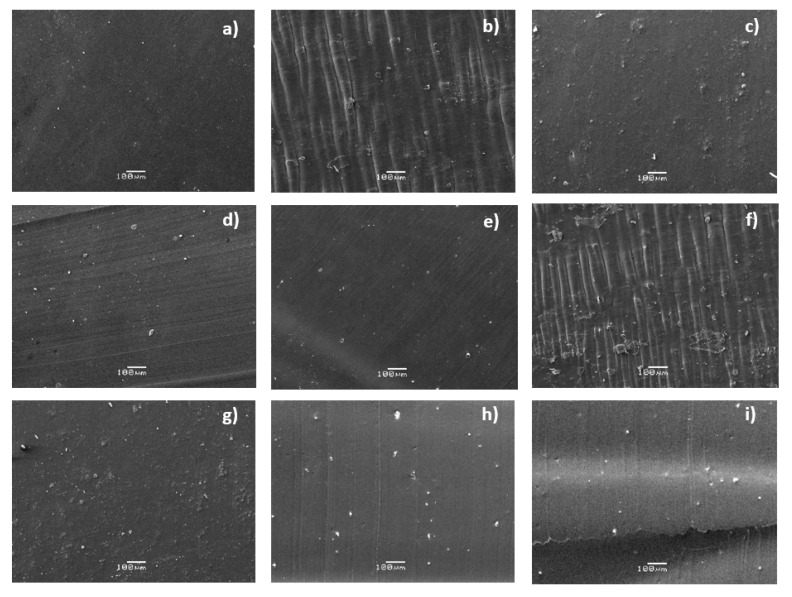
SEM micrographs of neat TPU and CPC materials: (**a**) Neat polyurethane; (**b**) 0.05 µm; (**c**) 0.2 µm; (**d**) 0.5 nm; (**e**) 1 µm; (**f**) 5 µm; (**g**) 10 µm;(**h**) 15 µm; (**i**) 20 µm.

**Figure 4 polymers-12-01934-f004:**
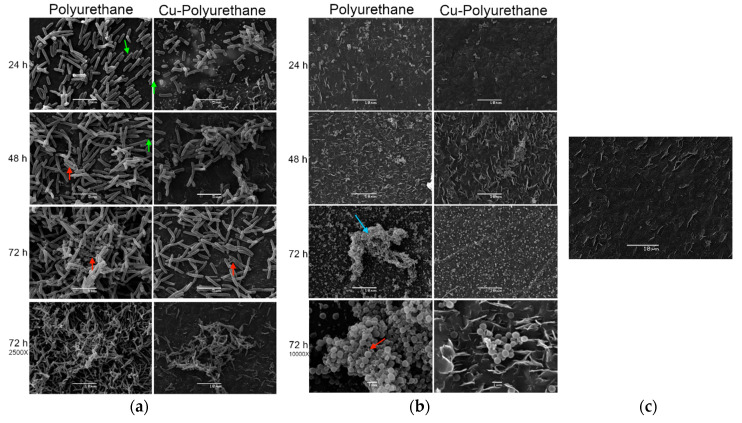
Biofilm formation on a copper-polyurethane composite of (**a**) *Escherichia coli* ATCC 25922, (**b**) *Staphylococcus aureus* ATCC 25923, and (**c**) a negative control (materials in tryptic soy broth (TSB) with no inoculum).

**Figure 5 polymers-12-01934-f005:**
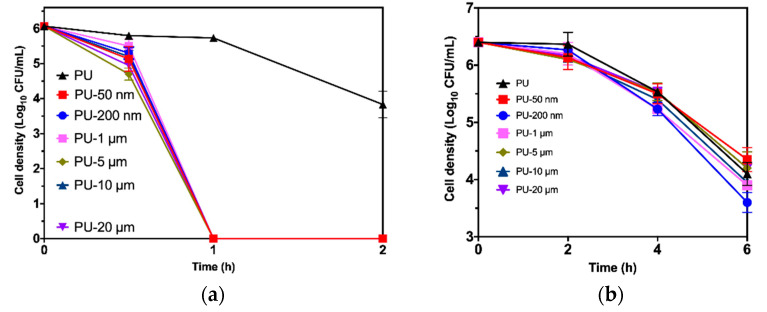
Bacterial death kinetics exhibited by copper-polyurethane composites: (**a**) *E. coli ATCC 25922*; (**b**) *S. aureus ATCC 25923*; PU was used as control.

**Figure 6 polymers-12-01934-f006:**
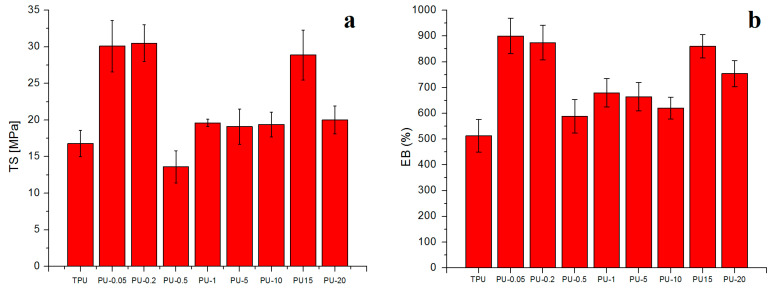
Mechanical properties obtained from the traction test of all CPC samples: (**a**) tensile strength, (**b**) elongation at break and (**c**) tensile modulus.

**Figure 7 polymers-12-01934-f007:**
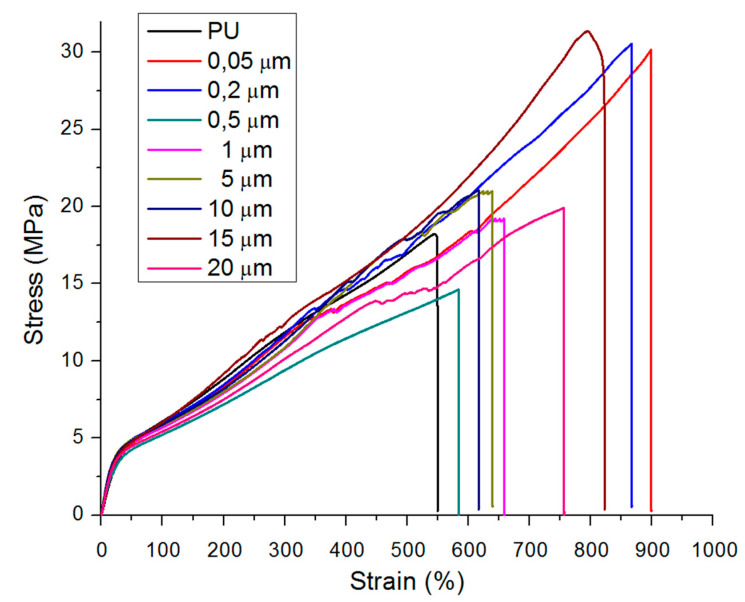
Representative tensile stress versus strain of the prepared sample with different particle sizes (tensile rate of 10 mm/min).

**Table 1 polymers-12-01934-t001:** Sample compositions and processability properties.

Sample Code	Poliurethane (wt%)	Copper Particle Size (µm)	Torque Max (Nm)	Torque Steady State (Nm)	MFI (g/10 min)
PU	100	0	106.9 ± 3.0	24.1 ± 0.32	4.1 ± 0.26
PU-0.05	99	0.05	101.7 ± 2.3	12.9 ± 0.31	9.1 ± 0.28
PU-0.2	99	0.2	99.6 ± 4.4	12.2 ± 0.15	8.2 ± 0.39
PU 0.5	99	0.5	100.8 ± 0.34	11.0 ± 0.23	6.7 ± 0.65
PU-1	99	1	111.0 ± 5.3	15.7 ± 0.10	6.8 ± 0.33
PU-5	99	5	111.2 ± 2.1	15.3 ± 0.19	7.7 ± 0.25
PU-10	99	10	108.8 ± 5.0	14.6 ± 0.07	11.0 ± 0.28
PU-15	99	15	113.5 ± 3.8	17.9 ± 0.26	8.1 ± 0.31
PU-20	99	20	111.8 ± 6.9	16.4 ± 0.08	8.2 ± 1.5

**Table 2 polymers-12-01934-t002:** Thermal properties of neat TPU and copper-polyurethane composite (CPC) materials.

	Initial Decomposition			Second Step Decomposition			Third Step Decomposition			
Sample Code	T_(max)_ (°C)	Loss Weight (%)	Degrad. Rate (%/min)	T_(max)_ (°C)	Loss Weight (%)	Degrad. Rate (%/min)	T_(max)_ (°C)	Loss Weight (%)	Degrad. Rate (%/min)	Residual (%)
PU	354.2	25.4	7.1	418.2	65.2	14.4				9.4
PU-0.05	366.2	27.3	6.7	421.5	64.2	12.5				8.5
PU-0.2	366.5	29.8	6.7	421.8	63.7	13.0				6.5
PU-0.5	362.9	8.9	5.9	374.5	33.5	3.4	426.5	50.1	11.8	7.5
PU-1	296.1	7.9	4.1	358.9	28.6	4.1	424.4	55.5	12.0	8.0
PU-5	258.2	6.7	3.7	346.6	24.9	3.7	428.5	53.8	12.8	8.0
PU-10	258.8	6.6	4.4	344.8	23.6	3.2	430.6	59.2	13.1	10.6
PU-15	338.9	15.9	6.5	361.9	31.1	3.2	420.3	42.4	12.2	10.6
PU-20	313.8	10.8	6.3	359.3	30.1	4.1	422.1	48.9	10.9	10.2

**Table 3 polymers-12-01934-t003:** Copper concentration in deionized water and physiological serum measured by ICP-OES.

	Cu^2+^ Concentration (µg/L) Deionized Water		Cu^2+^ Concentration (µg/L) Physiological Serum	
**Time (Days)**	**PU**	**1% Cu**	**PU**	**1% Cu**
0	<0.02	<0.02	<0.02	<0.02
0.05	<0.02	<0.02	<0.02	0.06
0.25	<0.02	<0.02	<0.02	0.39
1	<0.02	0.24	<0.02	1.51
2	<0.02	0.27	<0.02	2.43
3	<0.02	0.33	<0.02	2.95
7	<0.02	0.70	<0.02	3.83
14	<0.02	1.26	<0.02	5.44
21	<0.02	1.40	<0.02	5.94

**Table 4 polymers-12-01934-t004:** Effect of the copper size particles on the tensile properties.

Sample Code	E (MPa)	sM (MPa)	EB (%)	Integral Value (KJ/m^3^) *
PU	2.5 ± 0.2	16.8 ± 1.8	512.7 ± 63.4	2264.9
PU-0.05	2.9 ± 0.2	30.1 ± 3.5	900.2 ± 69.4	4092.4
PU-0.2	3.0 ± 0.3	30.5 ± 2.5	874.4 ± 67.2	4107.1
PU 0.5	1.7 ± 0.2	13.6 ± 2.2	589.4 ± 65.1	1831.2
PU-1	2.1 ± 0.1	19.6 ± 0.5	679.8 ± 54.5	2233.6
PU-5	2.4 ± 0.3	19.1 ± 2.4	664.7 ± 54.7	2563.2
PU-10	2.3 ± 0.2	19.4 ± 1.7	620.4 ± 42.6	2613.8
PU-15	2.2 ± 0.3	29.88 ± 3.4	859.9 ± 45.4	4006.2
PU-20	2.1 ± 0.1	20.0 ± 1.9	754.4 ± 50.5	2561.5

* Integral results of the stress–strain curves shown in Figure 7.

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
