# Peer review of "Copper-Polyurethane Composite Materials: Particle Size Effect on the Physical-Chemical and Antibacterial Properties"

_polymers, 2020, doi:10.3390/polym12091934_

Round 1

Reviewer 1 Report

The article (Polymers-857962) focuses on the effect of particle size copper nanoparticle on the properties of TPU composites. The results shown that the nano-size of copper can help to improve the properties of TPU, comparing with the micro-size particles, which having a potential biomedical application. The topic and the results are suitable to be published in POLYMERS. However, for the benefit of the reader, a series of revision are still suggested by this reviewer.

  1. Author mentioned “The purpose of this paper is to obtain by melt blending copper- thermoplastic polyurethane …” (L72). It is obviously wrong. Please revise it.
  2. Section 2.2, in the torque test using the torque rheometer, the torque value of steady state (balance torque) is actually difficult to be determined. How to determine the value, please clarify it.
  3. In section 2.3.3, the JSM 6380 LV microscope was repeated twice, is it necessary? The EDS, TSB should be defined before their using. In addition, are authors sure that the coated film has a 40nm thickness? It is also very strange that “the degree of distribution of Cu particles in the PVC matrix”.
  4. In L140~L142, the expression 2Log10, 3Log10 are really very strange.
  5. The pictures included has only four pictures. It is relatively poor. In fact, it is necessary to add more pictures to improve this paper’s impression. In this reviewer’s opinion, the curve of torque over time, the TGA curve, and the stress-strain curve are necessary to be illustrated.
  6. All the tested results have not given the errors. Actually, this reviewer believes that the experiments carried out in this study should have the errors. Please add or revise these, include table 1, table 2, and figure 4.
  7. Is there any possibility to analyze the results shown in Figure 1 and Figure 2 quantitatively, for example, the size of the Cu particles, the distance between the particles?
  8. It is interesting that both the tensile strength and elongation at break can be improved by the existence of 1wt% nano-copper. Could authors discuss and explain it more? A similar report that the nano-SiO2 was added into PP/LDPE, doi: 10.1177/1847980417715929, this reviewer recommends the author to read it and decide whether it is possibly to cite it for help reader to understand the reinforcing effect of nano-particles.
  9. A careful edition of language seems to be also necessary.

Author Response

Dear Reviewer

The answers point by point of your suggests and comments about the manuscript titled "Copper-polyurethane composite materials: Particle size effect on the physical-chemical and antibacterial properties" and corrected manuscript version, you can find in the attached file.

Dr. Cristian Miranda and Dr. Johanna Castaño
corresponding authors

Reviewer 2 Report

The topic of the paper is in line with the Journal’s aims and scopes. However, there is need for major revisions before it can be taken into account for publication.

Some comments:

1) In general, the manuscript should be revised and checked for typos, spelling, inconsistencies, inaccuracies, possibly with the help of a native speaker.

2) It is important that the Authors clearly explain what the novelty in this paper is, and especially the differences with other similar, recent papers.

3) Page 3, line 112: PVC matrix??

4) Table 1: “Stead state”?? furthermore, SI units must be clearly reported.

5) It is inaccurate to report micrometers as “um”. The suitable Greek letter should be used instead of “u”.

6) The use of torque values is a somewhat rough way to investigate the actual processability (flow properties). It is strongly recommended to perform suitable rheological characterization by using a rotational rheometer (over a wide frequency range) or a capillary rheometer.

7) Page 5, line 182: the “ball bearing” analogy is quite odd. Maybe it is better just to talk about lubricating effect by the nanoparticles,  or just about them decreasing the viscosity. Relevant literature should be cited and may help in using the most suitable words.

8) Mechanical characterization and discussion is poor. It must be improved. For instance, I could not find any table or graph regarding the elastic modulus. There are no error bars or any other information regarding the data scattering. There is no adequate discussion about the non-monotonic behavior of TS and EB on increasing the particle size. The trends of both TS and EB and their relationships with the particle size are not trivial at all, and thus need to be further discussed.

Author Response

(The authors gave the same response as above.)
